# IMPROVING AND EVALUATING OPEN DEEP RESEARCH AGENTS

## ABSTRACT

Deep Research Agents (DRAs) are systems that can take a natural language prompt from a user, and then autonomously search for, and utilize, internet-based content to address the prompt. Recent DRAs have demonstrated impressive capabilities on public benchmarks. However, recent research largely focuses on proprietary closed-source systems. At the time of this work, we identified only one open-source DRA, termed Open Deep Research (ODR). In this work, we adapt BrowseComp, the challenging recent benchmark dataset, to compare ODR to existing proprietary systems. We propose BrowseComp-Small (BC-Small), comprising a subset of BrowseComp, as a more computationally-tractable DRA benchmark for academic labs. We benchmark ODR and two other proprietary systems on BC-Small: one system from Anthropic and one system from Google. We find that all three systems achieve 0% accuracy on the test set of 60 questions. We introduce three strategic improvements to ODR, resulting in the ODR+ model, which achieves a state-of-the-art 10% success rate on BC-Small among both closed-source and open-source systems. We report ablation studies indicating that all three of our improvements contributed to the success of ODR+.

## 1 INTRODUCTION

In this work, we focus on the problem of developing Deep Research Agents (DRAs), wherein our goal is to develop a system that can take as input a natural language prompt from a user, then autonomously search for and utilize internet-based content to address the prompt. This is a challenging problem because, in principle, it typically comprises several sub-problems that are each challenging for contemporary artificial intelligence methods: for example, breaking a natural language prompt into easier sub-questions, reasoning about the use of an internet search engine to find relevant information on the internet, and then reasoning about that retrieved content to address the original prompt. Recently however, large language models (LLMs) have demonstrated the potential to address many of these challenges and several organizations have developed proprietary systems that seek to perform Deep Research. Examples include OpenAI's recent Deep Research OpenAI (2025b), Google's Deep Research Dave Citron (2024), and Perplexity's research capabilities Perplexity AI (2025).

One challenge with LLM-based DRAs is evaluating their performance, because the problems should, ideally, simultaneously satisfy two major competing properties. First, the problems must be sufficiently challenging so that they cannot already be easily solved by existing methods, such as a single prompt to an LLM, or a simple single query to a browser. Some existing benchmarks are theoretically suitable for DRAs, such as HotpotQA Yang et al. (2018) and Natural Questions Kwiatkowski et al. (2019), however recent LLM-based methods have achieved near-perfect accuracy, motivating the need for more challenging benchmarks. The increasing difficulty of the benchmark problems however makes it difficult to satisfy the 2nd needed property: any benchmark question should also include ground truth solutions to enable performance evaluation. Furthermore, it should be possible to find these solutions on the internet, or else they cannot be solved by a DRA. Therefore DRA benchmark problems must simultaneously be so difficult that their solutions are difficult to find, even by a human, but we must also be certain that there is a solution, and that it can be found on the internet.

Very recently, the BrowseComp OpenAI (2025a) benchmark was introduced to address the limitations of existing benchmarks. BrowseComp includes over 1200 problems that are, by design,

challenging to solve both for humans and existing LLM-based systems, while also being highly likely to have solutions that can be found via internet search. The authors of BrowseComp OpenAI (2025a) benchmarked several proprietary systems from OpenAI, and found that all of the systems (except one) performed poorly, achieving less than 10% accuracy. The very best system, which utilized specialized methods, and substantial test-time compute, was able to achieve 50% accuracy. One major limitation of the existing evaluation of BrowseComp is that it has so far focused entirely on proprietary, closed-source DRAs from OpenAI. This creates a limited picture of DRA capabilities, as open-source systems have not yet been systematically benchmarked. In practice, this is due in part to the high computational cost of running BrowseComp at scale, which has so far restricted thorough evaluation to organizations with access to substantial computational resources. At the time of this work, there was only one open-source DRA, termed Open Deep Research (ODR) Camara (2025). However, the performance of ODR has yet to be quantitatively evaluated, making it unclear how open source DRAs compare with proprietary counterparts, and there are currently no baseline methods upon which to improve DRAs within the open research community.

**Contributions of this work.** To address these limitations, we propose BrowseComp-Small, a more computationally tractable deep research benchmark, comprising two disjoint sets of sixty questions sampled from BrowseComp: a training set, intended for DRA development; and a testing set, intended for DRA performance evaluation. We found that ODR is unable to answer any of the challenging questions in the BrowseComp-Small testing set. We propose several methodological improvements to the ODR system to support more effective deep research, resulting in our proposed ODR+ system. ODR+ successfully answers 20% of the training and 10% of the test BrowseComp questions, and, therefore, greatly outperforms the original ODR system. We also find that ODR+ outperforms several proprietary closed-source systems as well. Using ablation studies on the BrowseComp benchmark, we demonstrate the benefit of each of our proposed methodological improvements to ODR. We provide an open-source implementation of ODR+ [1]. We summarize our contributions as follows:

1. We present one of the first quantitative benchmarks of open (or closed) DRAs, and the first such benchmark on the challenging recent BrowseComp benchmark.

2. We introduce ODR+, an open-source DRA that achieves state-of-the-art performance on the BrowseComp benchmark among open-source DRAs. We release the code for ODR+ to support continued progress.

3. We present ablation studies that provide evidence of the effectiveness of our individual proposed methodological innovations, providing insights to the community on building more effective DRAs. We also present a failure mode analysis, providing insight on why ODR+ fails, and thereby supporting the development of future open DRAs.

**We wish to note** that there is rapid progress on DRAs, and several open DRA benchmarks were published very recently, during the course of our work; we discuss these in Sec 2. The remainder of this paper is organized as follows: Sec. 2 discusses related work; Sec. 3 discusses the BrowseComp benchmark; Sec. 4 discusses ODR+, including a review of the original ODR system; Sec. 5 details our methodology; Sec. 6 discusses our experimental results; and Sec. 7 discusses our conclusions.

## 2 RELATED WORK

**Deep Research Agents.** Early work in autonomous deep research began with WebGPT (2021) Nakano et al. (2021) was the first piece of autonomous deep research work that enabled LLMs to ask actual Bing search questions and cite the results. It was unsuitable for benchmarking here, though, because it only supported single-turn QA and was not made available as a reusable browsing agent. Recent empirical benchmarking has confirmed that recent proprietary systems represent a significant improvement in apparent capability (discussed below). Prominent instances of real-time search and multi-step information retrieval include Google Gemini Deep Research, Perplexity AI, and OpenAI's Deep Research. The need for open, analyzable alternatives is prompted by the fact that these systems are closed-source despite their remarkable performance.

---

[1] URL of ODR+ implementation will be provided upon publication

**Existing Open Deep Research Agents.** A growing number of open-source systems have recently emerged with the goal of replicating the capabilities of proprietary deep research agents. To our knowledge, the earliest and most relevant example is Open Deep Research (ODR), which was limited due to its lack of benchmarking, and therefore motivated our work here. During preparation of our work, very recently, several other open deep research agents have been published.

DeepResearcher Zheng et al. (2025) introduced a reinforcement learning frameåwork for training browsing agents that autonomously decide what to search, read, and extract from the web. Another recent system is WebThinker Li et al. (2025), proposed a modular architecture to interleave deep Web exploration with reasoning, focusing on scientific and factual question answering. Although both provide code bases, at the time of our experiments they were not straightforward to reproduce; DeepResearcher required reinforcement-learning training runs with substantial GPU resources, and WebThinker's modular pipeline involved custom integration steps that were not yet fully documented or packaged for reuse. Given our limited compute budget and the fact that these systems had only just been released (e.g., and lacked full documentation) we were unable to obtain reliable runs suitable for benchmarking. We therefore restricted our comparisons to ODR, ODR+, and proprietary systems for which evaluation was feasible.

**Existing Deep Research Standards.** One of the main challenges for Deep Research agents is multi-hop reasoning across sources, where a *hop* is a single step of reasoning that links one piece of evidence to another or a question to a piece of evidence. HotpotQA Yang et al. (2018) and 2Wiki-MultiHopQA Ho et al. (2020) are benchmarks that focus on 2-hop questions, but they do so in a closed Wikipedia environment without open-ended search or query revision. Therefore, they do not adequately assess whether agents are capable of planning multi-step retrieval, generating queries on their own, or determining when additional evidence is required. The BrowseComp benchmark OpenAI (2025a), on the other hand, is a better test for Deep Research agents because it is made for complex multi-hop QA, which calls for conducting numerous web searches, obtaining a variety of evidence, and combining information from different documents.

In addition to BrowseComp, other benchmarks have very recently been developed and advanced the evaluation of DRAs. These benchmarks were released too recently for us to consider or include in our work, however, we describe them here to account for important related progress in this fast-paced research area. Mind2Web2 Gou et al. (2025) introduces a "agent-as-a-judge" framework for self-assessment and consists of 130 long-horizon tasks that require agents to browse unknown websites and generate structured, cited answers. However, the benchmark was published too recently to be included in our study and places more emphasis on self-assessment than on complex information synthesis. Though it covers fewer domains than BrowseComp and prioritizes factual verification over multi-hop reasoning, Deep Research Bench Huang et al. (2025a) consists of 89 live web tasks with reference answers and explicit evaluation criteria. A taxonomy and analysis of recent DRA benchmarks and system designs are also provided by Huang et al. Huang et al. (2025b).

## 3    THE BROWSECOMP-SMALL BENCHMARK

We first introduce the BrowseComp (BC) benchmark, from which we create the BrowseComp-Small (BC-Small) benchmark used in our experiments. **BrowseComp. OpenAI (2025a)** is a benchmark for Deep Research agents comprising 1,266 questions on a wide range of topics, including entertainment, science, history, politics, and geography. BrowseComp questions were constructed so that they are challenging to solve, even for humans, but have short answers that are easy to verify. Questions were constructed by human "trainers" using the following procedure. Each question was constructed by first identifying some object (e.g., a person, place, or thing), and then selecting a set of properties about that object that would, collectively, uniquely identify it. Using the selected set of properties, the designer would construct a question that asks the DRA to identify the object that satisfies all of the selected properties. For example, one BrowseComp question asks: *"Which 90s TV series starred an actor born in Tennessee, an actor who was a Caribbean immigrant, and an actor whose father was a law enforcement officer for more than 3 decades? The series was short-lived."*. The authors of BrowseComp used various criteria to ensure the difficulty of each question. For example, human trainers ensured that each question could not be correctly answered by another person within ten minutes. They also confirmed that existing models such as ChatGPT (with and

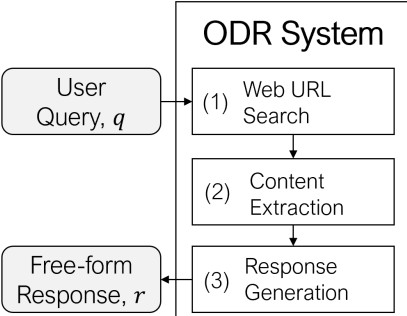

Figure 1: ODR system architecture illustrating the three main steps: (1) Web URL Search; (2) Content Extraction; and (3) Response Generation.

without browsing) and an early version of OpenAI's Deep Research were unable to solve them. See OpenAI (2025a) for full design details.

**BrowseComp-Small (BC-Small)** The computational cost of evaluating DRAs on the full BrowseComp benchmark is large, especially for academic labs. To make BrowseComp more accessible while maintaining its utility as a benchmark, we created a smaller benchmark - termed BrowseComp-Small (BC-Small) - that comprises a subset of 120 questions from BrowseComp. We sampled questions to maintain a similar distribution of topics as the full BrowseComp benchmark. Crucially, and in contrast to BrowseComp, we split BC-Small into 60 questions that are used for DRA development - essentially a training set - and another disjoint 60 questions as a testing set, with the goal of better evaluating DRA generalization. Our choice of 120 questions was chosen to be consistent with the size of other recent public benchmarks, such as Deep Research Bench Huang et al. (2025a) (89 questions) and Mind2Web2 Gou et al. (2025) (130 questions), where the authors also cited the high cost of issuing multiple search queries, parsing content, and performing iterative reasoning for each example.

## 4 OPEN DEEP RESEARCH

Here we describe the ODR system from Camara (2025), upon which our proposed ODR+ is based. We provide the essential system-level details of its operation, but further implementation details can be found in the supplemental information. An open-source implementation of ODR is also available[2].

The operation of ODR is illustrated in Fig. 1 and consists of three main steps. **(1) Web URL Search.** The user submits a natural language question, which is passed to a large language model (LLM) along with a system prompt instructing it to generate a concise search query suitable for an internet search engine. The LLM produces a single general-purpose query without decomposing the question or performing more reasoning. **(2) Content Extraction.** ODR submits the generated query to a search engine and retrieves a list of candidate web pages. It opens the top-ranked link and gets the rendered (i.e., visible to humans) content of the page, which is then converted to plain text without any extra filtering or parsing. (3) Generating a response. The LLM gets the extracted text and the original user question, along with a prompt instructing it to use only the retrieved content to generate an answer. The model gives the user a free-form answer, which is a natural language answer that doesn't have to follow a specific format.

## 5 OPEN DEEP RESEARCH PLUS (ODR+)

Here we describe our system, ODR+, which is constructed by making several improvements to the ODR system. ODR provides an important initial working system, but it suffers from several limitations that cause it to fail on complex, multi-hop research questions. We hypothesize that ODR

---

[2]ODR Implementation: https://github.com/nickscamara/open-deep-research.

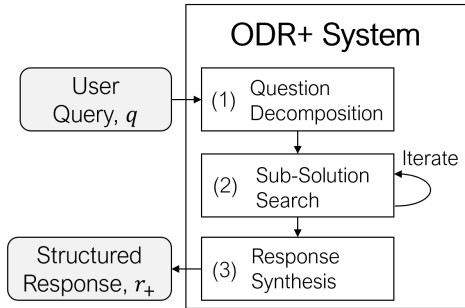

Figure 2: ODR+ system architecture, illustrating three major steps: (1) Question decomposition; (2) an iterative *Sub-solution Search* step, which seeks internet-based evidence to address each sub-question; and (3) *Response Synthesis*, where a structured response, denoted $r_+$, is generated for the user based upon a summary of evidence from the internet.

| Prompt | Prompt Name | Description |
|---|---|---|
| $P_1$ | Constraint Extraction | Asks the LLM to extract the specific constraints (e.g., names, dates, descriptors) from the user query to guide downstream reasoning. |
| $P_2$ | Sub-question Generation | Instruct the LLM to reformulate the original query into focused sub-questions that preserve key constraints. |
| $P_3$ | Content Extraction | Direct the LLM to extract only facts from retrieved web content that match specified constraints and current sub-question. |
| $P_4$ | Evidence Analysis | Ask the LLM to evaluate current findings, determine sub-question completion, and propose new sub-questions or termination. |
| $P_5$ | Response Synthesis | Instruct the LLM to aggregate all findings and output a structured final answer with confidence and justification. |

Table 1: Summary of engineered prompts used in ODR+ system, ordered by execution sequence. Full prompt text is included in the supplementary material.

fails on these problems for at least three reasons: it does not decompose the user query into simpler sub-questions; it lacks any form of iterative reasoning or adaptive planning; and it is not prompted to produce structured output. ODR+ addresses these limitations through the introduction of three modules illustrated in Fig 2: Question Decomposition, Sub-solution Search, and Response Synthesis. We next describe each of these three major modules, as well as sub-modules that contribute to them, which are detailed in pseudocode for ODR+ in Algorithm 1. The engineered prompts used in the ODR+ pseudocode are *summarized* in Table 1, and the full prompts are provided in the supplement.

## 5.1 QUESTION DECOMPOSITION

The first module of ODR+ converts the original user query, $userQuery$, into a set of focused sub-questions, as detailed in lines 7-8 of Algorithm 1. This begins with a call to `extractConstraints` (Line 7), which takes as input the prompt $P_1$ in Table 1 and $userQuery$. The prompt instructs the language model to extract explicit identifying details—such as names, dates, locations, or numerical values—that help narrow the search space. The output is returned in a simple structured format (e.g., a JSON list of constraints). For example, given the query *"Which 90s TV series starred an actor born in Tennessee and an actor who was a Caribbean immigrant?"*, the model would extract constraints like `["1990s","actor born in Tennessee","Caribbean immigrant"]`.

Next, the system calls `generateSubQuestions` (Line 8), which receives $P_2$, and $userQuery$, and the extracted constraints from $P_1$ above. Prompt $P_2$ guides the model to generate a small number of clear, fact-based sub-questions that target the extracted constraints. The resulting sub-questions are stored in the queue `S.subquestions`, which forms part of the system's internal research

state `S`. This state also tracks retrieved evidence, depth of search, processed URLs, and intermediate results, as initialized in Lines 4–5.

## 5.2 ITERATIVE SUB-SOLUTION SEARCH

This module focuses on addressing each of the sub-questions identified in module (1) and is shown in lines 10–30 of Algorithm 1. The iterative process continues until all sub-questions are addressed, or some other stopping criteria is met (e.g., permissible run-time, denoted $T_{max}$, is exceeded; or a maximum number of sub-questions, denoted $D_{max}$ is exceeded).

At the beginning of each iteration, the system selects an unresolved sub-question from `S.subquestions` (Line 12) and uses it directly as a web search query. We observed that web browsers returned a different page ranking each time the same query was submitted and therefore we submitted the same query $N_{query}$ times using `webSearch` (Line 13). The top-ranked URLs are gathered from each of the $N_{query}$ searches. The $k$ most frequently occurring URLs are then chosen for additional processing using `selectMostFrequent` (Line 14) after the frequency of each URL across all attempts is totaled.

Then, by calling `createExtractionPrompt` with the prompt template $P_3$, $userQuery$, and the constraints that were previously extracted in module (1), an extraction prompt `extractionPrompt` is created (Line 15). It is intended to give the LLM instructions to extract only the parts of the page content that are relevant to addressing the sub-question. The *extraction-Prompt* is passed to an LLM, along with the full text of each selected URL. The LLM is invoked once per URL and typically returns one, or a few, short spans of relevant text. These outputs are stored as structured findings—each consisting of the extracted text and its source URL—and are appended to `S.findings`, the list of accumulated findings maintained in the internal research state (Line 17).

After collecting new findings, ODR+ invokes an LLM using `analyzeEvidence` with prompt $P_4$, the current sub-question, and the full set of accumulated findings (Line 18). The prompt directs the model to generate a structured response in JSON format, which includes fields like a confidence score, a list of satisfied constraints, a proposed answer to the sub-question (if one can be found), and any recommended follow-up sub-questions. A valid response is added to `S.subAnswers` (Lines 19–20) following the parsing of the JSON output. If follow-up subquestions are suggested, they are added to `S.subquestions` (Lines 21–22). The model's analysis, particularly the confidence score and recommendation on whether to proceed, is also used by the control flow logic to decide whether to proceed to the next iteration or terminate the loop early (Lines 23–24).

## 5.3 RESPONSE SYNTHESIS

The third and final module of the ODR+ system, which is implemented in lines 32–33 of Algorithm 1, is responsible for synthesizing the final structured answer. The system uses the engineered prompt $P_5$ to invoke an LLM on line 32. It also includes the original user question (`userQuery`), the extracted constraints (`constraints`), and the accumulated evidence (`S.findings`). The prompt specifically prohibits reliance on prior knowledge and directs the model to produce a final response based only on this structured content. The model is asked to produce a response in the standardized BrowseComp format:

*Explanation:* {*reasoning based on findings*}
*Exact Answer:* {*short final answer or 'Unknown'*}
*Confidence:* {*confidence score as a percentage*}

The prompt also instructs the model to compute a confidence score based on the number of key constraints satisfied by the proposed answer, relative to the total number of extracted constraints. On line 33, the model's output is stored in `structuredResponse`. The system then validates this response to ensure that all required fields are present and correctly formatted. If any field (such as the explanation, exact answer, or confidence score) is missing or malformed, fallback values are inserted. For example, the system may assign `"Unknown"` as the answer and a default confidence score of 10%. This validation step ensures that every final output is complete, properly structured, and ready for automated evaluation.

---

**Algorithm 1** Open Deep Research Plus (ODR+)

---

1: **Input:** $userQuery$ (original user question)
2: **Output:** $structuredResponse$ (formatted final answer)
3: {Module 1: Question Decomposition}
4: **Initialize:**
5: $S \leftarrow \{findings : [], depth : 0, processedUrls : \emptyset,$
    $urlFreqMap : \{\}, subquestions : [], subAnswers : [],$
    $timeLimit : T_{max}, maxDepth : D_{max}\}$
6: $t \leftarrow 0, startTime$
7: $constraints \leftarrow$ extractConstraints$(P_1, userQuery)$
8: $S.subquestions \leftarrow$
    generateSubQuestions$(P_2, userQuery, constraints)$
9: {Module 2: Iterative Sub-Solution Search}
10: **while** $S.depth < D_{max}$ **and** $t < T_{max}$ **and**
    $S.subquestions \neq \emptyset$ **do**
11:    $S.depth \leftarrow S.depth + 1$
12:    $currentSubQuestion \leftarrow S.subquestions$.pop()
13:    $urls \leftarrow$ webSearch$(currentSubQuestion, N_{query})$
14:    $topUrls \leftarrow$ selectMostFrequent$(urls, k)$
15:    $extractionPrompt \leftarrow$
      createExtractionPrompt$(P_3, userQuery, constraints)$
16:    $newFindings \leftarrow$ extractFromUrls$(topUrls, extractionPrompt)$
17:    $S.findings \leftarrow S.findings \cup newFindings$
18:    $analysis \leftarrow$
      analyzeEvidence$(P_4, S.findings, currentSubQuestion)$
19:    **if** $analysis.subAnswer \neq null$ **then**
20:      $S.subAnswers \leftarrow S.subAnswers \cup \{analysis.subAnswer\}$
21:    **end if**
22:    **if** $analysis.subquestions \neq \emptyset$ **then**
23:      $S.subquestions \leftarrow$
      $S.subquestions \cup analysis.subquestions$
24:    **end if**
25:    **if** $(analysis.hasAnswer \wedge analysis.confidence \neq low)$ **or**
     $\neg analysis.shouldContinue$ **then**
26:      **break**
27:    **end if**
28:    wait$(W_{ms})$
29:    $t \leftarrow$ getCurrentTime$() - startTime$
30: **end while**
31: {Module 3: Response Synthesis}
32: $structuredResponse \leftarrow$
    synthesizeResponse$(P_5,$
      $userQuery, constraints,$ S.findings$)$
33: **return** $structuredResponse$

---

## 6 NUMERICAL EXPERIMENTS

We conduct experiments on our BrowseComp-Small benchmark (see Sec. 3), which comprises two disjoint sets of sixty questions: a training set and a testing set. We evaluate several competing DRA systems on the sixty test questions: ODR, ODR+, Claude-DeepResearch (Anthropic), and Gemini-DeepResearch (Google 2025).

### 6.1 ODR+ DEVELOPMENT AND HYPERPARAMETER SETTINGS

All development of ODR+ was done using the sixty training questions in our BrowseComp-Small benchmark. This was done to minimize the potential of overfitting the design of ODR+ to the testing questions. Many steps of ODR+ (and ODR) utilize an LLM, and we utilized the **GPT-4o-mini** model via the OpenAI API. This model was selected because it allows for scalable evaluation under constrained compute budgets and offers a good trade-off between cost, latency, and reasoning quality. For ODR+, we used the following hyperparameter settings:

- **Search Depth** ($D_{\max} = 6$): The system performs up to six iterative search hops per question.

- **Time Limit** ($T_{\max} = 210$ seconds): Each question must complete within 3.5 minutes of wall-clock time.

- **Top-$k$ URLs** ($k = 3$): At each hop, the system selects the $k$ most frequent URLs across multiple search attempts.

- **Search Retries** ($N_{query} = 3$): Each sub-question is submitted to the search engine $N_{query}$ times to reduce variability in returned results.

These hyperparameters were chosen through experimentation on the training set, balancing answer quality, runtime, and the cost of running the model. We note however that increasing these hyperparameter settings may likely improve system accuracy, at the cost of increased computational cost — we did not have the resources to investigate this potentiality.

## 6.2 EVALUATION METHODOLOGY

We follow the official BrowseComp evaluation protocol, which requires system responses to conform to a standardized three-part structure (Explanation, Exact Answer, Confidence). Each system output is scored using the released BrowseComp evaluator, which leverages the **GPT-4o** model (via OpenAI API) to assess both answer correctness and formatting adherence. The evaluator performs semantic comparison between the predicted answer and the ground truth to determine exact match accuracy. Therefore, for each question the evaluator determines whether the response of the DRA is correct or incorrect, and we report the resulting accuracy over the 60 test questions of each system. All web searches and page extractions in ODR and ODR+ were performed using FireCrawl to ensure consistent and structured retrieval.

## 6.3 MAIN RESULTS

The main results are reported in Table 2. ODR was unable to answer any questions in the BrowseComp-Small test set, whereas ODR+ answered **10%** (6 of 60) with exact-match correctness. In BrowseComp, "exact match" is determined by the official evaluator, which requires structured responses (Explanation, Exact Answer, Confidence) and uses GPT-4o to check semantic equivalence with the ground truth. Because BrowseComp answers are short (e.g., names, numbers, or short phrases), this evaluation is highly reliable. *To our knowledge, ODR+ achieves the current state-of-the-art (SOTA) performance on the BrowseComp benchmark among open-source models.*

Surprisingly, ODR+ also outperformed the two proprietary DRAs we tested: Claude-DR and Gemini-DR, both of which achieved 0% accuracy on the 60-question test set. Because these systems do not expose structured outputs, we manually reviewed their answers against the ground truth. In nearly all cases, their outputs were long, report-style responses rather than the short exact answers required by the benchmark. We inspected these generated report, and confirmed that they did not contain the correct answers, so their accuracy remained 0%. We note that ODR+ was developed using a separate 60-question training split, whereas ODR, Claude-DR, and Gemini-DR were evaluated zero-shot on the test set, introducing a potentially significant disadvantage for them. Unfortunately at the time of our experimentation, these proprietary systems could not be tuned or adjusted for a custom benchmark such as BrowseComp. Our experiments represent our best attempt to evaluate them fairly and transparently, however, our methods still imposed the aforementioned disadvantages.

For completeness, we also report the results of ChatGPT-DR that were reported in OpenAI (2025a), which were obtained on the full BrowseComp benchmark, and which varied depending on test-time compute, from $\sim$10% with limited compute to 51.5% with extensive compute. The paper shows performance scaling with browsing effort and sampling, but does not specify the exact compute allocations for these settings, making direct comparison difficult. Unlike our setup, ChatGPT-DR was potentially developed using the entire BrowseComp benchmark rather than a disjoint train/test split, which may provide an advantage.

In addition to accuracy, we also measured average wall-clock runtime. ODR+ required $\sim$198 seconds per question, close to its fixed 210 s limit. This time limit kept bounded compute, and ODR+ typically used the full available budget. By contrast, ODR failed to complete runs, while Claude-DR averaged 11 minutes and Gemini-DR 4 minutes per question under default APIs. Thus, ODR+'s performance cannot be attributed to greater compute availability, since proprietary systems actually consumed more time on average.

**Multi-Judge Validation.** To validate our LLM-based evaluation methodology, we re-evaluated all 60 ODR+ test responses using five independent judges spanning two major AI providers: OpenAI (GPT-4o, GPT-4o-mini, GPT-3.5-turbo) and Google (Gemini 2.5 Flash, Gemini 2.5 Pro). All judges reached perfect agreement (Cohen's $\kappa = 1.00$) on every question, each independently scoring

Table 2: Performance and Runtime Comparison on BC-Small Test Set

| Deep Research Agent | LLM | Accuracy (%) | Avg. Runtime / Q |
|---|---|---|---|
| ODR | GPT-4o-mini | 0% | N/A |
| ODR+ (ours) | GPT-4o-mini | 10% | 198s |
| Claude-DR | Sonnet 4 | 0% | 11 min |
| Gemini-DR | Gemini 2.5 Pro | 0% | 4 min |
| ChatGPT-DR | GPT-4o | ~10–51.5%* | N/A |

*Results reported from OpenAI (2025a) on the full BrowseComp benchmark.

6/60 correct (10.0% accuracy). This strong agreement across both vendors and model architectures demonstrates that LLM-based evaluation is highly reliable for BrowseComp's factual questions. Complete details are provided in the supplementary materials.

**Performance Stability.** We also conducted experiments aimed at demonstrating robustness of ODR+'s results to two major sources of randomness: randomness in (i) the testing data, and (ii) in the ODR+ algorithm itself. Note that we cannot test robustness to the training data because ODR+ has no trained parameters, and most baseline models are closed-source. To test (i) we created an independent 60-question test set using identical stratified sampling as BC-Small, with zero overlap with the BC-Small train and test questions. We then ran ODR+ on these new questions and obtained 11.67% accuracy (7 of 60 correct), compared to 10% on the original test set. This suggests that the performance of ODR+ is robust to randomness in the testing dataset. We also ran the ODR and Gemini-DR baselines on this new test set. Gemini-DR and it achieved 6.67% correct (4 of 60), compared to 0% (0 of 60) on the original test set. ODR achieved 0% correct on both test sets. These results indicate that the performance advantage of ODR+ persists on the new dataset, and is unlikely to have occurred by chance. Furthermore, the modest improvement in Gemini-DR is not necessarily due to randomness; because Gemini-DR is a closed-source system, it may have been updated/improved since it was last tested (e.g., the Gemini LLM used in Gemini-DR appears to have been updated from 2.5 to 3.0 since our previous experiments).

To address (ii) we ran ODR+ three times on our new 60-question test set to observe its consistency across these trials. The results were as follows: Run 1 achieved 7 of 60 correct (11.67%); Run 2 achieved 5 of 60 correct (8.33%); and Run 3 achieved 6/60 (10.00%), with overall accuracy of 18/180 = 10.00% (SD = 1.70%). The 95% confidence interval [7.5%, 12.5%] was computed using bootstrap resampling from the three independent runs. Run-to-run consistency was 93% (56/60 questions identical outcomes). These results suggest robustness to randomness in the ODR+ algorithm as well. Further analysis is provided in the supplementary materials.

**Failure Mode Analysis.** We systematically analyzed failures by comparing retrieved web content to ground truth answers. We found 85% of correct answers were not in any retrieved pages, reflecting BrowseComp's puzzle-like design with multiple rare constraints. Proprietary systems (Claude-DR, Gemini-DR) with unlimited time also achieved 0% accuracy, demonstrating inherent difficulty. Full details are provided in the supplementary materials.

## 6.4 ABLATION STUDIES

To understand the impact of individual components in ODR+, we conducted ablation studies by disabling key modules and observing performance changes. Due to computational costs, we randomly selected 20 test questions from the BC-Small test benchmark and evaluated the following ablated variants of ODR+:

- **No Sub-question Decomposition:** The sub-question generation and decomposition step is disabled in this variant. The system returns to the original ODR's single-query methodology. On multi-hop questions, which usually call for breaking down complex prompts into more manageable, targeted searches, we anticipate a notable decline in performance.

- **No Iterative Planning:** Adaptive planning and research state management are eliminated in this variant. Sub-questions are handled one after the other without the use of retry logic

or feedback based on past results. This restricts the system's ability to dynamically modify its approach, which probably lowers the efficiency of information gathering.

- **No Structured Synthesis:** This variant eliminates structured output formatting and validation. It uses free-text generation like the original ODR instead. We expect lower estimates of confidence, formatting problems, and a higher chance of getting final answers that are wrong or incomplete.

Table 3 shows how disabling each core module reduces the accuracy of the ODR+ system, highlighting the overall contribution of each component to system effectiveness.

Table 3: Ablation Study Results on 20-Question Subset

| System Variant | Accuracy (%) |
|---|---|
| ODR+ | 25% (5/20) |
| No Structured Synthesis | 0% (0/20) |
| No Sub-question Decomposition | 5% (1/20) |
| No Iterative Planning | 5% (1/20) |

## 7 CONCLUSIONS

We introduced ODR+, an enhanced open-source Deep Research Agent (DRA) designed to perform complex multi-hop web-based question answering. Building on the original ODR system - the only open-source DRA we could identify at the outset of this research - ODR+ incorporates several improvements: sub-question decomposition, iterative planning, and structured synthesis. We benchmarked ODR+ on the BrowseComp-Small dataset, a subset of the BrowseComp benchmark that we curated for more scalable DRA benchmarking, and demonstrated that it significantly outperforms the original ODR baseline, achieving 10% exact-match accuracy on the test set while producing answers in the required format (Explanation, Exact Answer, Confidence). We also present evidence that ODR+ is competitive with proprietary systems, although fair comparisons are difficult. Our ablation studies confirmed the critical role of our three proposed improvements over ODR. To support continued progress in the development and evaluation of DRAs, we release our implementation and tools publicly. We hope ODR+ serves as a foundation for future research in open, analyzable, and extensible Deep Research Agents.

## REPRODUCIBILITY STATEMENT

We have made an effort to make sure our results can be reproduced. We give a full description of our curated BrowseComp-Small benchmark in Section 3, including train/test split. We also explain the exact architectures of both the original ODR and enhanced ODR+ systems in Sections 4 and 5. To help make things clearer, we provide pseudocode (Algorithm 1) and prompt summaries (Table 1). Section 6 and the supplementary materials have full details on how to set hyperparameters and the evaluation methodology, along with complete implementation details ensuring all the experiments can be replicated. Finally, we provide ablation studies (Sec. 6.4) to show the contribution of each system component.

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
