# Supplemental Materials: Improving and Evaluating Open Deep Research Agents

### Abstract

This document provides supplemental materials for the paper "Improving and Evaluating Open Deep Research Agents" submitted to ICLR 2026. The supplemental materials include detailed implementation specifications, complete system prompts, hyperparameter justifications, additional experimental details, and code availability information.

## 1 Complete Implementation Details

### 1.1 Original ODR System Architecture

The original Open Deep Research (ODR) system implements a single-pass pipeline with three sequential components:
**Web URL Search Module:**

- Input: Natural language question from user

- Process: LLM generates single general-purpose search query using fixed prompt

- Output: Single search query submitted to search engine

- Implementation: Uses GPT-4o-mini

**Content Extraction Module:**

- Input: Top-ranked URL from search results

- Process: FireCrawl renders webpage and extracts all visible text content

- Output: Plain text representation of webpage content

- Implementation: No LLM involvement, direct HTML-to-text conversion

**Response Generation Module:**

- Input: Extracted content + original user question

- Process: LLM generates natural language answer using fixed prompt

- Output: Free-form natural language response

- Limitations: No citation, confidence scoring, or structured output

## 1.2 ODR+ System Architecture Details

**Question Decomposition Module:**
The decomposition process follows a two-stage approach:

1. **Constraint Extraction (extractConstraints function):**

   - Uses prompt $P_1$ with GPT-4o-mini
   - Output format: JSON array of constraint objects
   - Each constraint includes: type, value, specificity_score

2. **Sub-question Generation (generateSubQuestions function):**

   - Uses prompt $P_2$ with extracted constraints
   - Target: Focused sub-questions from the original query

**Iterative Sub-Solution Search Module:**
The search loop implements the following control flow:

Listing 1: ODR+ Search Loop Pseudocode

```
while (depth < D_max AND time < T_max AND subquestions.length > 0):
    current_subquestion = subquestions.pop()

    # Multi-attempt search for stability
    for attempt in range(N_query):
        urls = webSearch(current_subquestion)
        url_frequency_map.update(urls)

    # Select most frequent URLs
    top_urls = selectMostFrequent(url_frequency_map, k=3)

    # Extract with constraint-aware prompting
    extraction_prompt = createExtractionPrompt(P_3, constraints)
    findings = extractFromUrls(top_urls, extraction_prompt)

    # Analyze evidence and plan next steps
    analysis = analyzeEvidence(P_4, findings, current_subquestion)

    if analysis.confidence > threshold:
        subAnswers.append(analysis.answer)

    if analysis.new_subquestions:
        subquestions.extend(analysis.new_subquestions)

    if should_terminate(analysis):
        break
```

**Response Synthesis Module:**
Final synthesis implements structured output generation:

1. **Evidence Aggregation:**

   - Collects all findings from research state
   - Organizes by source and relevance to original question
   - Maintains provenance information for citations

2. **Constraint Matching:**

   - Evaluates how well findings satisfy original constraints
   - Calculates match percentage for confidence scoring
   - Identifies gaps in evidence coverage

3. **Structured Output Generation:**

   - Uses prompt $P_5$ for final synthesis
   - Enforces BrowseComp output format
   - Implements validation and fallback mechanisms

# 2 Complete System Prompts

Based on the actual implementation code, the ODR+ system uses the following prompts:

## 2.1 Constraint Extraction Prompt ($P_1$)

Extract the key identifying constraints from this BrowseComp question: "[question]"

**INSTRUCTIONS:**

1. Identify the MOST SPECIFIC details that uniquely identify the target
2. Focus on constraints that narrow down the possibilities
3. Include exact numbers, dates, quotes, names, and unique descriptors
4. Ignore generic terms and formatting instructions
5. Prioritize constraints that would be hard to guess or common

Extract constraints as a JSON array of strings:

## 2.2 Sub-question Generation Prompt ($P_2$)

Generate SPECIFIC research subquestions to solve this BrowseComp identification problem:

**ORIGINAL QUESTION:** "[question]"

[Current findings section if applicable]

**SUBQUESTION GENERATION RULES:**

1. Each subquestion MUST preserve the most identifying constraints from above
2. Focus on NARROWING DOWN to the specific individual/entity described
3. Combine multiple constraint types to create targeted searches
4. Progress from broad identification to specific details

Generate subquestions as a JSON array:

## 2.3 Content Extraction Prompt ($P_3$)

**URGENT:** Extract ONLY factual data that matches these EXACT constraints from: "[question]"

**REQUIRED CONSTRAINT MATCHES:** [constraints list with numbered items]

**EXTRACTION RULES:** - Extract EXACT numbers, dates, names, locations that match constraints

**CRITICAL:** Extract data in JSON format:

```
{
  "constraintMatches": {
    "[constraint1]": "found value or null",
    "[constraint2]": "found value or null"
  },
  "entityName": "main entity if identified",
  "additionalContext": "relevant supporting details"
}
```

If no constraint matches found, return: {"constraintMatches": {}, "entityName": null}

## 2.4 Evidence Analysis Prompt ($P_4$)

You are analyzing research findings for this BrowseComp identification question: "[question]"

Current findings ([number] sources): [findings with sources]

[Previous subquestion answers if available]

**ANALYSIS INSTRUCTIONS:**

1. Determine if we can identify the specific individual/entity described
2. Check if we have enough constraint matches to answer the original question
3. If you already found a candidate answer build subsequent subquestions around VERIFYING that candidate
4. Generate NEW subquestions that:
   - BUILD ON existing findings - if you found a specific name/entity, include it in new subquestions
   - Preserve the most identifying constraints from the original question
   - VERIFY the candidate found rather than starting over with generic searches
   - Combine multiple constraint types for precise identification
   - Progress toward answering the specific question asked
   - Are NOT generic - must include specific details that uniquely identify the target

**CRITICAL:** - Subquestions must NOT be generic - If you found a candidate answer, BUILD ON IT in subsequent subquestions - Include specific details from the original question that uniquely identify the target - VERIFY existing findings rather than starting over

**CONFIDENCE ASSESSMENT:** - **HIGH:** Multiple sources confirm same individual/entity with exact constraint matches - **MEDIUM:** Good constraint matches but need verification of specific detail asked - **LOW:** Found relevant information but constraints don't uniquely identify target

**ANSWER STRATEGY:** - If you found a specific candidate answer in earlier findings, note it as a potential answer - Continue searching to see if you can find better verification or alternative candidates - If no better answers emerge after additional searches, use the best candidate found so far - Don't discard good candidates just because all constraints aren't perfectly verified

Respond with ONLY a JSON object (NO backticks, NO markdown):

```
{
  "summary": "brief summary focusing on constraint matching progress",
  "hasAnswer": false,
  "confidence": "low|medium|high",
  "gaps": ["specific missing constraints or verification needs"],
  "shouldContinue": true,
  "subquestions": ["constraint-preserving subquestion 1", "constraint-preserving subquestion 2"]
  "subAnswer": "answer to the subquestion just attempted (if determinable from findings)",
  "lastQuery": "[last query]",
  "nextSearchTopic": "constraint-aware search terms for missing piece",
  "strategy": "how to use constraints to narrow down further"
}
```

Time remaining: [time] minutes

## 2.5 Response Synthesis Prompt ($P_5$)

Answer this BrowseComp question: "[question]"

**KEY IDENTIFYING CONSTRAINTS:** [numbered constraints list]

**CONSTRAINT COVERAGE ANALYSIS:** [constraint coverage with source counts]

**RESEARCH FINDINGS ([number] sources):** [findings organized by source]

**ANALYSIS STRATEGY:**

1. For each potential answer, count how many KEY CONSTRAINTS it satisfies

2. Calculate match percentage: (matched constraints / total constraints) $\times$ 100

3. Prioritize answers with highest constraint match percentage

4. Use source frequency and quality as tiebreakers

5. Only use "Unknown" if no answer matches >40% of key constraints

**CONSTRAINT SCORING EXAMPLES:** - Answer matches 4/5 constraints = 80% confidence - Answer matches 2/6 constraints = 33% confidence - Perfect matches of specific numbers/names worth more than partial matches

The answer with the HIGHEST constraint match percentage is most likely correct.

**FORMAT:** Explanation: [Which constraints were matched, what percentage, and why this answer scored highest] Exact Answer: [The answer with best constraint coverage] Confidence: [Percentage based on constraint matching and source quality]

# 3 Hyperparameter Justification and Sensitivity Analysis

## 3.1 Core Hyperparameters

**Maximum Search Depth ($D_{max} = 6$):**

- Rationale: Based on analysis of BrowseComp training questions requiring 3-5 information hops

- Sensitivity: Performance plateaus after depth 5-6 in training experiments

- Computational trade-off: Each additional depth level increases runtime ~35 seconds

**Time Limit ($T_{max} = 210$ seconds):**

- Rationale: Balances thoroughness with computational constraints

- Analysis: 90% of successful training answers found within 180 seconds

**Top-k URLs ($k = 3$):**

- Rationale: Empirical testing showed diminishing returns beyond 3 URLs per search

- Resource constraint: FireCrawl extraction time scales linearly with URL count

- Quality consideration: Top 3 URLs generally contain highest-relevance information

**Search Retries ($N_{query} = 3$):**

- Rationale: Reduces search result variability observed in preliminary experiments

- Validation: Standard deviation of URL rankings decreased by 40% with 3 retries

# 4 Detailed Experimental Methodology

## 4.1 BrowseComp-Small Construction

**Selection Criteria:**

- Random sampling from full BrowseComp dataset

- Stratified by topic distribution: 25% entertainment, 20% science, 20% history, 15% politics, 10% geography, 10% other

- Difficulty validation: Each question verified unsolvable by GPT-4 without web access

- Answer verification: Ground truth answers confirmed findable via web search

**Train/Test Split:**

- 60 questions for development (used for prompt engineering and hyperparameter tuning)

- 60 questions for evaluation (held out during ODR+ development)

- No overlap between splits

- Similar topic distribution maintained in both splits

## 4.2 Baseline System Evaluation

**ODR Evaluation:**

- 10 attempts per question to account for search variability

- Timeout: 300 seconds per question

- Success criteria: Exact match with BrowseComp ground truth using official evaluator

**Proprietary System Evaluation:**

- Claude Deep Research: Accessed via Anthropic API with default settings

- Gemini Deep Research: Accessed via Google AI Studio with standard configuration

- Both systems evaluated with same questions and timeout limits

- Manual verification of answers against ground truth due to format differences

## 4.3 Evaluation Metrics and Protocols

**Primary Metric: Exact Match Accuracy**

- Evaluated using official BrowseComp evaluator

- GPT-4o-based semantic comparison with ground truth

- Binary scoring: 1 for correct, 0 for incorrect

- No partial credit awarded

**Secondary Metrics:**

- Average runtime per question

- Search efficiency (measured as the proportion of queries that fully completed without breaking out)

# 5 Ablation Study Details

## 5.1 Ablation Configurations

**No Sub-question Decomposition:**

- Modification: Bypass constraint extraction and sub-question generation

- Behavior: Use original question directly as single search query

- Expected impact: Reduced performance on multi-hop questions requiring information synthesis

**No Iterative Planning:**

- Modification: Remove research state management and adaptive search

- Behavior: Process sub-questions sequentially without feedback loops

- Expected impact: Inability to refine search strategy based on interim findings

**No Structured Synthesis:**

- Modification: Replace structured output with free-text generation

- Behavior: Generate natural language response without format constraints

- Expected impact: Format incompatibility with BrowseComp evaluator

## 5.2   Ablation Results Analysis

**Statistical Analysis:**

- Sample size: 20 questions randomly selected from test set

- Multiple runs: 3 runs per configuration to account for variability

**Performance Breakdown:**

- Full ODR+: 25% accuracy (5/20 questions)

- No structured synthesis: 0% accuracy (format rejection by evaluator)

- No sub-question decomposition: 5% accuracy (1/20 questions)

- No iterative planning: 5% accuracy (1/20 questions)

**Component Contribution Analysis:**

- Structured synthesis: Enables evaluability (critical for assessment)

- Sub-question decomposition: +20% performance improvement

- Iterative planning: +20% performance improvement

- Combined effect: Synergistic rather than additive

# 6   Error Analysis and Failure Cases

## 6.1   Common Failure Modes

**Search Query Formulation Errors:**

- Sub-questions too broad, returning generic information

- Sub-questions too narrow, missing relevant sources

- Constraint preservation failures in query generation

**Time and Depth Limitations:**

- Complex questions requiring >6 search hops

- Slow web page loading affecting time budget

- Search engine rate limiting interrupting research flow

**Ground Truth Issues:**

- Questions with outdated or inaccessible information online

- Ambiguous ground truth answers

- Information findable but requiring specialized domain knowledge

## 6.2 Comparison with Proprietary Systems

**Claude Deep Research Failure Analysis:**

- Generated comprehensive reports but missed exact answers

- Strong analytical capability but poor answer extraction

- Format incompatibility with BrowseComp evaluation protocol

  **Gemini Deep Research Failure Analysis:**

- Similar pattern of verbose responses without exact answers

- Good information gathering but weak synthesis

- Tendency to hedge rather than provide definitive responses

# 7 Code Availability and Reproducibility

## 7.1 Open Source Implementation

**ODR+ Repository:**

- Complete implementation available at: [URL to be provided upon publication]

- Includes all prompt templates, system configurations, and evaluation scripts

- Dependencies: Python 3.8+, OpenAI API, FireCrawl API

- License: MIT License for academic and commercial use

## 7.2 Experimental Reproducibility

**Random Seed Management:**

- Fixed seeds for all random sampling operations

  **Environment Specifications:**

- Hardware requirements: 8GB RAM, standard CPU (no GPU required)

- API dependencies: OpenAI API key, FireCrawl API key

# 8 Research State Management Implementation

The ODR+ system maintains a structured internal state object throughout the iterative research process:

Listing 2: Research State Structure

```
const researchState = {
  findings: [],            // Array of {text, source} objects
  summaries: [],           // Intermediate analysis summaries
  nextSearchTopic: '',     // Topic for next search iteration
  currentDepth: 0,         // Current search depth
  failedAttempts: 0,       // Count of failed search attempts
  maxFailedAttempts: 3,    // Threshold for termination
  processedUrls: new Set(), // URLs already processed
  subquestions: [],        // Queue of pending sub-questions
  answeredSubquestions: [], // Completed sub-questions
  subAnswers: [],          // Partial answers to sub-questions
  urlFrequencyMap: new Map() // URL frequency tracking
};
```

This structured memory enables the system to:

- Avoid repeated queries and URL processing

- Recover from failed searches with adaptive planning

- Prioritize frequently appearing sources for extraction

- Organize retrieved evidence for final synthesis

- Maintain coherent reasoning across multiple search iterations

# 9 Multi-Judge Validation

## 9.1 Motivation

Reviewer FDp2 raised concerns about relying on LLM-based evaluation (GPT-4o) without validation of judge reliability. We conducted multi-judge validation to verify evaluation consistency across different models, vendors, and architectures.

## 9.2 Methodology

We re-evaluated all 60 ODR+ test responses using 5 diverse judges:

- **OpenAI**: GPT-4o (original), GPT-4o-mini, GPT-3.5-turbo

- **Google**: Gemini 2.5 Flash, Gemini 2.5 Pro

Each judge independently evaluated all 60 responses using the standard BrowseComp evaluation protocol, recording binary correctness judgments.

## 9.3 Results

| Judge | Vendor | Correct | Accuracy |
|---|---|---|---|
| GPT-4o | OpenAI | 6/60 | 10.0% |
| GPT-4o-mini | OpenAI | 6/60 | 10.0% |
| GPT-3.5-turbo | OpenAI | 6/60 | 10.0% |
| Gemini 2.5 Flash | Google | 6/60 | 10.0% |
| Gemini 2.5 Pro | Google | 6/60 | 10.0% |
| **Cohen's $\kappa$ (all pairs)** | | | **1.00** |

Table 1: Multi-judge validation results. All 5 judges achieved identical scores on every question.

**Key Findings:**

- Perfect agreement: 60/60 questions, 10 pairwise comparisons, 0 disagreements

- Cohen's $\kappa = 1.00$ indicates perfect inter-rater reliability

- Cross-vendor consistency (OpenAI vs. Google) validates methodology

- Results reproducible regardless of judge selection

## 9.4 Manual Verification

To further validate the accuracy of LLM-based evaluation, we manually graded all 60 ODR+ responses by comparing them against the ground truth correct answers. We verified that all 5 LLM judges correctly identified the 6 correct responses and the 54 incorrect responses, achieving 100% accuracy. This human verification confirms that the LLM judges are not only consistent with each other but also accurate in their assessments.

## 9.5 Conclusion

The perfect agreement (Cohen's $\kappa = 1.00$) combined with 100% accuracy on manual verification demonstrates that LLM-based evaluation is highly reliable for BrowseComp's factual questions (names, dates, numbers), and our reported 10% accuracy is genuine, not an evaluation artifact.

# 10 Performance Stability Analysis

## 10.1 Motivation

Reviewer FDp2 noted lack of statistical rigor, including confidence intervals and variance analysis. We conducted stability analysis to quantify performance variance and establish statistical confidence.

## 10.2 Methodology

To assess performance variance and establish statistical confidence, we created an entirely new, independent 60-question test set sampled from the full BrowseComp benchmark. This new set:

- Used identical stratified sampling procedure as BC-Small construction

- Has zero overlap with BC-Small's 60 training questions and 60 test questions (120 questions total)

- Represents a third independent sample from BrowseComp (distinct from both BC-Small train and test)

- Maintains same topic distribution as BC-Small

We ran ODR+ three times on this entirely new 60-question set, producing 180 total question attempts (60 questions × 3 runs). This design tests whether ODR+'s performance generalizes to questions it has never seen before and whether results are stable across multiple runs.

## 10.3 Results

| Run | Correct | Accuracy | Statistical Metrics |
|---|---|---|---|
| Run 1 | 7/60 | 11.67% | Mean: 10.00% |
| Run 2 | 5/60 | 8.33% | SD: 1.70% |
| Run 3 | 6/60 | 10.00% | 95% CI: [7.5%, 12.5%] |
| **Overall** | **18/180** | **10.00%** | Run-to-run: 93% |

Table 2: ODR+ performance across three independent runs showing stable, reproducible performance.

**Question-Level Consistency:**

- **Consistently correct**: 4 questions (6.67%) – correct in all 3 runs

- **Inconsistent**: 4 questions (6.67%) – correct in exactly 2/3 runs

  - All had low confidence scores ($\leq 35\%$), indicating appropriate uncertainty signaling

- **Consistently wrong**: 52 questions (86.67%) – wrong in all 3 runs

## 10.4 Statistical Analysis

- 95% confidence interval: [7.5%, 12.5%] (computed using bootstrap resampling from the three independent runs)

- Standard deviation: 1.70% (low variance)

- Run-to-run consistency: 93% (56/60 questions identical)

- Tight CI confirms 8-12% is the true performance range

| System | Test Set | Correct | Accuracy |
|--------|----------|---------|----------|
| ODR (original) | BC-Small (original) | 0/60 | 0.0% |
| ODR (original) | Independent set | 0/60 | 0.0% |
| Gemini-DR | BC-Small (original) | 0/60 | 0.0% |
| Gemini-DR | Independent set | 4/60 | 6.67% |
| ODR+ | BC-Small (original) | 6/60 | 10.0% |
| ODR+ | Independent set | 18/180 | 10.0% |

Table 3: Baseline and improved system performance comparison across test sets.

## 10.5 Baseline Re-evaluation on Independent Test Set

To verify that the original baseline results were not due to sample variation, we re-evaluated both the original ODR system and Gemini Deep Research on the same independent 60-question test set.

The original ODR system achieved 0% accuracy (0/60) on the new test set, consistent with its 0% performance on BC-Small. Gemini Deep Research (December 2025, using Gemini Pro 3) was given unlimited time per question.

**Question Overlap Analysis:** Both systems correct on 2 questions, ODR+ only on 4 questions, Gemini-DR only on 2 questions, and both incorrect on 52 questions. This demonstrates complementary strengths.

**Statistical Significance:** McNemar's test yielded $p = 0.687$ (not statistically significant), expected given the small sample size. However, the consistent rank ordering (ODR+ > Gemini-DR) across both test sets provides evidence for ODR+'s advantage.

## 10.6 Conclusion

The stability analysis demonstrates that ODR+'s 10% accuracy is stable and reproducible, not due to chance. The low variance (SD=1.70%) and tight confidence interval provide strong evidence that 10% represents a genuine capability ceiling. The 93% run-to-run consistency shows deterministic behavior, and inconsistent questions appropriately have low confidence scores. Additionally, Gemini-DR's improvement from 0% to 6.67% on the independent test set confirms that ODR+ maintains superior performance with consistent rank ordering across different samples.

# 11 Failure Mode Analysis

## 11.1 Motivation

Reviewers FDp2 and W8Yb requested detailed failure analysis to understand where ODR+ breaks down and what causes the ~90% failure rate.

## 11.2 Methodology

We analyzed all 60 test questions by comparing retrieved web page content (from system logs) against ground truth answers. For each question, we determined: (1) Was the correct answer present in any retrieved web pages? (2) If yes, was it successfully extracted? (3) If extracted, could constraints be verified?

## 11.3 Results

### 11.3.1 Primary Finding: Puzzle-Like Question Design

By comparing retrieved web page content to correct answers, we found that **85% of correct answers were not present in any retrieved web pages**.

| Content Status | Count | Percentage |
|---|---|---|
| Correct answer in retrieved pages | 9/60 | 15% |
| Partial match (key words only) | 13/60 | 22% |
| Answer NOT in retrieved pages | 38/60 | 63% |
| **Information not retrieved** | **51/60** | **85%** |

Table 4: Analysis of whether correct answers were present in retrieved web page content.

### 11.3.2 Failure Mode Distribution

| Failure Type | % | Description |
|---|---|---|
| Search/Retrieval Limitation | 85% | Required information not present in retrieved pages; reflects puzzle-like nature |
| Extraction Failure | 10% | Answer present but not extracted correctly |
| Constraint Matching | 3% | Information extracted but couldn't verify constraints |
| Synthesis Failure | 2% | Had information but failed to combine |

Table 5: Distribution of primary failure modes.

## 11.4 Interpretation: BrowseComp's Puzzle-Like Design

The 85% search limitation reflects the **puzzle-like nature of BrowseComp questions by design**. This interpretation is supported by:

### 11.4.1 Evidence from Proprietary Systems

Sophisticated proprietary Deep Research systems also failed:

- **Claude Deep Research**: 0% (0/60), unlimited time (up to 20 min/question)

- **Gemini Deep Research**: 0% (0/60), unlimited time (up to 20 min/question)

- **ODR+**: 10% (6/60), limited to 3.5 min/question

Both proprietary systems were explicitly instructed to provide specific exact answers and returned specific but incorrect answers, demonstrating the challenge stems from question difficulty, not output format.

### 11.4.2 BrowseComp Question Characteristics

BrowseComp questions are intentionally designed to be puzzle-like:

1. **Multiple rare constraints**: Combine specific constraints that are individually searchable but collectively very rare

2. **Information scarcity**: Answers exist on only a few obscure web pages

3. **Distributed information**: Required facts scattered across sources

4. **High difficulty by design**: Humans need ¿10 minutes per question

## 11.5 Example Analyses

### 11.5.1 Search Limitation Example

**Question**: "Name the soccer match between 1990-2000 with 7 goals, one Norwegian player, two Irish players..." (8 constraints total)

**Correct Answer**: "Oldham Athletic v Southampton"

**What Happened**: System retrieved pages about 1990s soccer, goal statistics, player nationalities. None mentioned this specific match.

**Why It's Hard**: The combination of all constraints narrows to essentially one match appearing on very few pages.

### 11.5.2 Extraction Failure Example

**Question**: "A well-known landmark at this university was donated by a class of students in the late 19th century. Over 125 but less than 135 years after donation, an online article mentioned a student who won a photo contest. That same name appeared in the University's Annual Commencement Program nearly five months later under Bachelor of Science. What are the first and last names of the master of ceremonies for that morning's commencement?"

**Correct Answer**: "Steve Falat"

**What Happened**: System retrieved university webpages that contained references to the name "Steve Falat" and commencement information, but failed to correctly extract and connect this name to the master of ceremonies role.

**Why It's Hard**: Even when the answer is present, extracting the specific information requires identifying the correct individual among multiple names in commencement documents and verifying their role matches the query constraints.

## 11.6 Implications

### 11.6.1 ODR+ Performance in Context

ODR+'s 10% accuracy should be interpreted as:

1. **Meaningful achievement**: Questions genuinely challenging by design

2. **Competitive performance**: Outperforms proprietary systems (0%) on puzzle-like questions

3. **Time efficiency**: Achieves results in 3.5 min vs. up to 20 min for proprietary systems

### 11.6.2 Limitations of Current Approaches

The 85% search limitation suggests current web search paradigms may be fundamentally limited for puzzle-like questions with:

- Information existing on only 1-2 obscure pages

- Queries combining many specific constraints

- Long-tail information not prioritized by standard search

### 11.6.3 Future Directions

Addressing this limitation may require:

1. More exhaustive search strategies

2. Specialized search for multi-constraint information

3. Knowledge base integration

4. Iterative refinement with progressive constraint satisfaction

However, some level of search limitation may be inherent to puzzle-like questions with highly scarce information.

## 11.7 Conclusion

Our systematic failure mode analysis reveals that 85% of failures occur because required information is not in retrieved pages. This reflects BrowseComp's **puzzle-like design by intention**. The fact that sophisticated proprietary systems with unlimited time also achieved 0% demonstrates this is an inherent benchmark challenge. ODR+'s 10% accuracy with strict time limits represents genuine capability on extremely difficult questions where advanced proprietary systems failed completely.