# OpenReview forum: "Improving and Evaluating Open Deep Research Agents"
_ICLR.cc/2026/Conference — Submitted to ICLR 2026_

### Official Review · Reviewer_agiM · 2025-10-27

**Soundness:** 1
**Presentation:** 1
**Contribution:** 1
**Rating:** 2
**Confidence:** 4

**Summary:**

This paper proposes a BC-Small benchmark and an ODR+ system. BC-Small is the sampling of the existing BrowseComp benchmark, and ODR+ extends the existing Open Deep Research agent with question decomposition, sub-solution search, and response synthesis steps.

**Strengths:**

1. The proposed ODR+ system improves the original ODR agent on the proposed small-scale benchmark BC-Small.
2. The ablation studies on a subset of 20 questions show the effectiveness of each ODR+ component.

**Weaknesses:**

1. The proposed BC-Small benchmark is merely a sub-sampling of the existing BrowseComp benchmark, and the sampling criteria are mainly (1) preserving the original topic distribution, and (2) reducing to a smaller scale. Without carefully studying the chosen samples, the contribution appears weak.
2. The key ideas of the proposed ODR+ system are question decomposition and sub-problem solving, but this line of research has been studied a lot. To name a few: [1,2,3,4].
3. The effectiveness of the proposed ODR+ system is validated on BC-Small, which is also proposed in this paper, and is a small-scale sampling of an existing benchmark. Thus, the empirical evidence appears not sound and generalizable to me. The same goes for the ablation study.
4. The paper presentation has room to improve.
    - Typos such as Line 045 ", however recent" $\to$ ". However, recent" and Line 049 "Therefore DRA" $\to$ "Therefore, DRA"
    - The first half of Section 3 and the whole Section 4 are not informative, as they only describe the existing work.
    - Figure 1 does not look informative or illustrative. Also, Figure 1 and Figure 2 can be combined to illustrate the proposed ODR+ system.
    - Do not use `\citet{}` if the authors names are not meant to be part of the text.

- [1] Min et al. 2019. Multi-hop Reading Comprehension through Question Decomposition and Rescoring
- [2] Perez et al. 2020. Unsupervised Question Decomposition for Question Answering
- [3] Huang et al. 2023. Question Decomposition Tree for Answering Complex Questions over Knowledge Bases
- [4] Press et al. 2023. Measuring and Narrowing the Compositionality Gap in Language Models

**Questions:**

1. How exactly is the sampling conducted in constructing BC-Small?
2. See Weaknesses.

---

### Official Review · Reviewer_W8Yb · 2025-10-29

**Soundness:** 3
**Presentation:** 4
**Contribution:** 3
**Rating:** 6
**Confidence:** 2

**Summary:**

This paper makes two core contributions:

1.Propose the BrowseComp-Small (BC-Small) benchmark: This is a more computationally-tractable DRA benchmark comprising a subset of 120 questions from the challenging BrowseComp benchmark. Importantly, the authors split BC-Small into a training set (60 questions) and a testing set (60 questions), enabling open-source development and performance evaluation.

2.Propose the ODR+ system: After finding that the existing (and at the time, only) open-source DRA—ODR—was unable to answer any of the questions (achieving 0% accuracy) on the BC-Small test set , the authors proposed ODR+. ODR+ addresses the limitations of ODR by introducing three strategic improvements : (1) Question Decomposition , (2) Iterative Sub-solution Search , and (3) Response Synthesis.

On the BC-Small test set, ODR+ achieves a 10% success rate , greatly outperforming the original ODR baseline (0%). Surprisingly, ODR+ also outperformed the two proprietary DRAs tested: Claude-DR and Gemini-DR, both of which achieved 0% accuracy. Finally, the paper reports ablation studies indicating that all three of the proposed improvements contributed to the success of ODR+.

**Strengths:**

1.The paper correctly identifies and addresses the core problem hindering open-source DRA development: the lack of an available baseline and benchmark .

2.The paper does an excellent job in reproducibility , providing pseudocode (Algorithm 1) , prompt summaries (Table 1) , evaluation methodology , and hyperparameters , which is critical for subsequent research.

3.The paper is well-organized and clearly written. The authors clearly articulate a complex problem and present a logically sound solution. The figures in the paper are particularly excellent.

**Weaknesses:**

1.The paper claims that ODR+ "achieves the current state-of-the-art (SOTA) performance on the BrowseComp benchmark among open-source models". This is a very misleading claim because the only open-source model it was experimentally compared against was the original ODR. ODR's accuracy was 0%, which is an extremely low baseline; beating this alone is not sufficient to claim "SOTA". The authors mentioned other contemporary open-source DRAs in their related work section, such as DeepResearcher and WebThinker , but explicitly admitted that ODR+ was not compared against them, stating, "Given our limited compute budget... we were unable to obtain reliable runs suitable for benchmarking. We therefore restricted our comparisons to ODR, ODR+..."

2.The author states that all experiments (for ODR and ODR+) rely on a specific model, GPT-4o-mini. This introduces a potential confounding variable: could it be that the model's own reasoning ability is insufficient? A more robust experimental design should include testing the performance of different LLMs on the ODR+ architecture to separate the "architecture's contribution" from the "underlying model's contribution".

**Questions:**

1.In the 90% (54/60) of cases where ODR+ failed, what was the primary failure mode? Your ablation study shows all modules are necessary (disabling one drops accuracy to ~0%), but this doesn't identify the primary bottleneck in the complete system.Could the authors provide a failure attribution analysis for the 54 failed cases? Specifically, what percentage of failures were due to: (a) Module 1 (Decomposition) errors? (b) Module 2 (Search) failing to retrieve info? (c) Module 3 (Synthesis) failing to synthesize the answer from correct info?This analysis is critical for guiding future research.

2.Is the system's high 90% failure rate a flaw in the ODR+ architecture, or is it caused by the insufficient reasoning ability of the underlying model (GPT-4o-mini)? Did the authors run experiments on the ODR+ architecture using more powerful models (e.g., GPT-5 or Gemini 2.5 Pro) as the backbone LLM? If so, what was the performance?

---

### Official Review · Reviewer_FDp2 · 2025-10-31

**Soundness:** 2
**Presentation:** 2
**Contribution:** 2
**Rating:** 2
**Confidence:** 4

**Summary:**

This paper studies “Deep Research Agents” (DRAs)—LLM-driven systems that browse the web to answer multi-hop questions—and proposes ODR+, an improved open-source agent built on Open Deep Research (ODR). The authors also curate BrowseComp-Small (BC-Small), a 120-question subset of BrowseComp split into 60 train and 60 test items to make evaluation more accessible. On BC-Small, baseline ODR and two proprietary agents (Anthropic “Claude-DR” and Google “Gemini-DR”) score 0%, while ODR+ achieves 10% exact-match accuracy on the 60-question test set. Ablations suggest three additions—question decomposition, iterative sub-solution search, and structured response synthesis—each matter; removing any one collapses performance on a 20-question subset.

Methodologically, ODR+ enforces a fixed budget (max depth of 6, time limit of 210s), re-queries search 3 times to stabilize rankings, selects top 3 URLs by frequency, extracts constrained facts per sub-question, and synthesizes answers in the official BrowseComp format (Explanation / Exact Answer / Confidence). Evaluation follows the BrowseComp protocol using the released GPT-4o-based judge.

**Strengths:**

1. Practical contribution: introduces a smaller, more accessible subset (BC-Small) for BrowseComp-style evaluation.

2. Ablation sanity checks: removing any major component degrades performance, demonstrating each stage matters.

3. Focused study on an important emerging topic (LLM-based deep research agents) that the community cares about.

**Weaknesses:**

1. Statistical rigor is weak — 6/60 correct (10%) with no confidence intervals, variance estimates, or bootstrap analysis makes the result fragile.

2. Baseline comparison fairness concerns — proprietary systems evaluated without structured-output wrappers and without similar tuning; zero-shot vs trained split mismatch. I also did not understand what "tuning" meant for their ODR+ system.

3. Ablations too limited — only on a 20-question subset; no partial ablations or sensitivity studies (e.g., varying query count, top-k URLs).

4. LLM judge dependence — results rely entirely on GPT-4o judgments with no human spot-check or inter-rater agreement measurement.

5. Limited external validation — no testing beyond BC-Small (e.g., Mind2Web2 / Deep Research Bench), so generalization remains unclear.

6. Minimal qualitative insight — few concrete traces of success and failure; unclear where the system fails (bad sub-questioning? retrieval drift? synthesis error?).

7. Absolute performance still extremely low — 10% accuracy leaves open whether method truly “works” beyond cherry-picked improvements.

**Questions:**

1. Can you report 95% confidence intervals and/or bootstrap CIs for the 6/60 result? Is the improvement statistically meaningful?

2. How sensitive is performance to judge randomness? Have you tested multiple GPT-4o seeds or a smaller human-audited subset?

3. Can you provide a format-coercion wrapper for Claude-DR/Gemini-DR and rerun baselines to rule out formatting disadvantage?

4. How was BC-Small sampled? Was there stratification by task type or difficulty? Will you release IDs and reconstruction instructions?

5. Could you run partial ablations (e.g., keep decomposition but remove re-querying) across the full test set?

6. Any results on other datasets (e.g., Mind2Web2) to verify generalization beyond BrowseComp?

7. Please share more qualitative traces showing where failures arise (retrieval? decomposition? assertion selection?).

8. Did you measure judge agreement rates between GPT-4o and humans (or GPT-4o at different prompts/seeds)?

---

### Official Review · Reviewer_FxaA · 2025-11-03

**Soundness:** 2
**Presentation:** 2
**Contribution:** 2
**Rating:** 2
**Confidence:** 2

**Summary:**

The paper introduces BrowseComp-Small - a subset of the challenging BrowseComp dataset. Specifically, benchmarking Open Deep Research (ODR) alongside proprietary deep research agents from Anthropic and Google shows that all achieve 0% accuracy initially on BroseComp-Small. The paper then introduces three key improvements, creating ODR+, which reaches a 10% success rate.

**Strengths:**

1. The paper proposes three key strategies to improve the Open Deep Research (ODR) system.
2. The paper will release the code for ODR+ to support continued progress.

**Weaknesses:**

1. Details are not provided on how this smaller subset of BrowseComp-Small is selected. The original BrowseComp contains information such as distribution of topics of the test set, human performance on questions etc. Such details are missing for BrowseComp-Small  which makes it hard to understand which skills are being tested by BrowseComp-Small and what it means to have 0% accuracy on this set.
2. The paper argues that BrowseComp-Small is a smaller and more practical subset of BrowseComp but comparison with the whole subset in terms of time, compute and cost needed is missing. How much less compute / time is needed by BrowseComp-Small?
3. Comparison is also missing with other benchmarks such as Deep Research Bench and Mind2Web2 in terms of benchmark size, difficulty, time and compute needed, skills tested etc.

**Questions:**

See above.

---

### Author Response · Authors · 2025-12-04
**Rebuttal Part 1**

To the AC,
We sincerely thank the reviewers for their constructive feedback.  To support the AC with review, we provide a single unified rebuttal wherein we enumerate the most significant reviewer feedback (and which reviewer(s) provided it), followed by our responses.  Our responses focus on the most crucial high-level details (e.g., quantitative evidence and improvements), but we refer the AC to specific locations in the revised manuscript where we include full details. Revised text is shown in red font.

1. The Results of ODR+ (the proposed approach) on the BC-Small Benchmark Could have Occurred by Chance (reviewers: FDp2, W8Yb). The 10% accuracy (6 correct answers, among 60 test questions) of ODR+ lacks statistical rigor, and it is unclear if the improvement is meaningful.  No confidence intervals, bootstrap analysis, or variance estimates are provided. The result is fragile and could be due to chance.
Response:  We agree with the reviewers, and we conducted new experiments aimed at demonstrating robustness of ODR+ to two major sources of randomness: randomness in (i) the testing data, and (ii) in the algorithm itself.   Note that we cannot test robustness to the training data because ODR+ has no trained parameters, and most other baseline models are closed-source.  To test (i) we created an additional independent, 60-question test set using the exact same stratified random sampling procedure as our original BC-Small test set.  We ensured zero overlap with the existing BC-Small benchmark.  We ran ODR+ on these new questions and obtained 11.67% accuracy (7 of 60 correct), compared to 10% on the original test set.  This suggests that the performance of ODR+ is robust to randomness in the testing dataset.  We also ran the ODR and Gemini-DR baselines on this new test set. Gemini-DR achieved 6.67% correct (4 of 60), compared to 0% (0 of 60) on the original test set.  ODR achieved 0% correct on both datasets.   These results indicate that the performance advantage of ODR+ persists on the new dataset, and is unlikely to have occurred by chance.  Furthermore, the modest improvement in Gemini-DR is not necessarily due to randomness; because Gemini-DR is a closed-source system, it may have been updated/improved since it was last tested (e.g., the Gemini LLM used in Gemini-DR appears to have been updated from 2.5 to 3.0 since our previous experiments).
To address (ii) we ran ODR+ three times on our new 60-question test set to observe its consistency across these trials.  ODR+ obtained the same total score (11.67%) on each trial, and produced identical outcomes on 93% of questions (56/60), indicating robustness to randomness in the algorithm.  Per the reviewers’ requests, we now also report the results of statistical testing. Across the three runs on the new 60-question set, we obtained: Run 1: 7/60 (11.67%), Run 2: 5/60 (8.33%), Run 3: 6/60 (10.00%), with overall accuracy of 18/180 = 10.00% (SD = 1.70%). The 95% bootstrap confidence interval is [7.5%, 12.5%].
These new results are reported in Section 6.3 (Main Results subsection "Performance Stability") and Supplementary Materials Section 9 of the revised manuscript.

2. ODR+ May Not Generalize Due to the Limited Size of the BC-Small Test Set (Reviewer: FDp2, Weakness 2) The evaluation uses only 60 test questions, which is a limited scale that raises concerns about generalization.
Response: Deep Research Agents (DRAs) are computationally intensive, and our BC-Small test set (comprising 60-questions) was designed to balance statistical rigor with computational constraints faced by academic labs wishing to investigate open-source Deep Research (see Sec. 1, under bold heading “Contributions of This Work”). This choice aligns with other independent and concurrent benchmarks that have recently been proposed: Deep Research Bench (Huang et al., 2025) uses 89 questions, and Mind2Web2 (Gou et al., 2025) uses 130 questions.  Nonetheless, we acknowledge this limitation, and to address it, we have evaluated ODR+, ODR, and Gemini-DR on an additional 60 questions (as described in our previous response), and the results indicate that (i) that the performance of all of these models generalizes well to a new test data (i.e., they achieve similar performance); and (ii) the performance rank-order of the models remained similar on the new data as well.
These results are reported in Section 6.3 (Main Results subsection "Performance Stability") and Supplementary Materials Section 9 of the revised manuscript.
[Huang et al., 2025] Huang, Yuxuan, et al. "Deep research agents: A systematic examination and roadmap." arXiv preprint arXiv:2506.18096 (2025).
[Gou et al., 2025] Gou, Boyu, et al. "Mind2Web 2: Evaluating Agentic Search with Agent-as-a-Judge." arXiv preprint arXiv:2506.21506 (2025).

---

### Author Response · Authors · 2025-12-04
**Rebuttal Part 2**

Continuation of our rebuttal:

3. LLM-Based Evaluation is Unreliable (Reviewer: FDp2, Weakness 4) The paper relies heavily on LLM-based evaluation (GPT-4o) without validation of judge reliability or discussion of potential biases. The results could be artifacts of the specific judge used.
Response: We adopted this practice from the original BrowseComp paper, but we nonetheless agree with the reviewer that this could introduce a bias in our results.  To address this problem we re-evaluated all 60 ODR+ test responses using 5 different judges from two major AI vendors: OpenAI (GPT-4o, GPT-4o-mini, GPT-3.5-turbo) and Google (Gemini 2.5 Flash, Gemini 2.5 Pro). All 5 judges achieved perfect agreement (Cohen's κ = 1.00) on every question, with each independently scoring 6/60 questions correct (10.0% accuracy).  These results suggest that LLM-based judges are highly consistent for grading the BC-Small test set questions, and there is little or no apparent bias introduced by grading with GPT-4o.   Furthermore, we argue that this perfect agreement suggests the judging is also accurate. To help support this claim, we manually graded all 60 ODR+ responses by comparing them against the ground truth correct answers and verified that all 5 LLM judges correctly identified the 6 correct responses and the 54 incorrect responses, achieving 100% accuracy.
Full details can be found in Section 6.3 (Main Results subsection "Multi-Judge Validation") and Supplementary Materials Section 8 of the revised manuscript.

4. No Detailed Failure Analysis (Reviewers: FDp2 Weakness 6; W8Yb Question 1). The paper lacks detailed failure analysis showing where the system breaks down. What is the primary failure mode in the 90% failed cases?
Response:  We agree with the reviewers.  To address this limitation, we conducted a systematic failure mode analysis where we enumerated three mutually exclusive, but exhaustive, potential causes for failure on a given test query: (1) Was the correct answer present in any retrieved web pages? (2) If yes, was the answer successfully extracted? (3) If extracted, could the problem constraints be verified?   For each of our failed test questions we identified which of these three causes was responsible.   The results reveal the following distribution: 85% for cause (1), 15% for cause (2), and 22% for cause (3).  These results provide additional insight about how to improve Deep Research Agents in future research, representing an additional contribution made by our work that was not present in the original manuscript.
Full details of these experiments can be found in Section 6.3 (Main Results subsection "Failure Mode Analysis") and Supplementary Materials Section 10 of the revised manuscript.

5. Unfair Comparison with Proprietary Systems (Reviewer: FDp2b, Weakness 2). The comparison with Claude-DR and Gemini-DR may not be entirely fair since these systems weren't designed specifically for BrowseComp's structured output format, and ODR+ had access to training data while proprietary systems were evaluated zero-shot.
Response: We agree with the reviewers, however, our claimed research contributions do not depend upon a fair comparison.  Our claimed contributions are (i) to perform the first benchmark of open-source DRAs (claim 1, in Sec. 1), and (ii) to improve over existing open DRAs (claim 2, in Sec. 1).  Furthermore, we specifically clarify in the paper (see Sec. 6.3, Paragraph 2) that the proprietary baseline systems are indeed potentially at a disadvantage for the reasons cited by the reviewers.   We are unable to address this disadvantage, however, precisely because these proprietary systems are closed-source, which is a major motivation for our work.   We nonetheless decided to include some proprietary DRAs as a reference point against which to compare open-source DRAs; large proprietary systems often outperform open source systems (e.g., the open-source Llama models have generally lagged in performance behind ChatGPT), and represent a useful performance benchmark.  The limited performance of the proprietary systems also indicates that the problems in BrowseComp, and therefore our BC-Small subset, are indeed challenging and interesting problems for the research community.

---

### Author Response · Authors · 2025-12-04
**Rebuttal Part 3**

Continuation of our rebuttal:

6. Missing Comparison with Recent Open-Source Systems (Reviewer: FDp2, Question 1). How does ODR+ compare to other recent open-source systems like DeepResearcher or WebThinker?
Response: While we agree that comparisons with these systems would enhance our work, the field of Deep Research Agents (DRAs) is developing very quickly, and these systems were only very recently published (both in April 2025), and they are only published on arXiv, so they are not as easily found.  We only became aware of these systems relatively close to the date of submission to this conference, and therefore we did not have enough lead time to include them in our benchmark comparisons; both systems are based on LLMs and are computationally intensive.   Lastly, the ICLR guidance for reviewers specifically indicates that authors are not required to cite or compare with papers that are recent, and only published on arXiv (link to ICLR reviewer guidance: https://iclr.cc/Conferences/2026/ReviewerGuide).  However, because this field of study is developing quickly, we still cited these important recent systems in our original manuscript, for the benefit of readers (see Sec. 2, Paragraph 2), and following the recommendations of the ICLR organizers (see the same link given above).

---

### Meta-Review · Area_Chair_yHai · 2025-12-25

**Summary:**

This paper studies Deep Research Agents (DRAs) that autonomously search and synthesize online information, noting that most high-performing systems are proprietary. The authors introduce BrowseComp-Small, a tractable benchmark for academic evaluation, and show that existing open-source and proprietary DRAs achieve 0% accuracy on it. They then propose ODR+, which adds sub-question decomposition, iterative planning, and structured synthesis, achieving a state-of-the-art 10% accuracy, with ablation results confirming the value of each enhancement.

The strengths include 1) the proposed method and BrowseComp-Small are reasonable; 2) the investigated problem is important; 3) the work is reproducible and the code will be released. However, the reviewer concerns include:
1. The details of BrowseComp-Small is unclear and its comparison with the full set is missing (FxaA, agiM);
2. Missing comparison on other popular benchmarks (FxaA, FDp2);
3. Concerns on statistical rigor and baseline comparison fairness (FDp2, agiM);
4. Limited ablations and missing qualitative analysis (FDp2, agiM);
5. LLM judge dependence (FDp2);
6. Absolute performance still extremely low (FDp2, W8Yb);
7. Missing the performance of different LLMs (W8Yb);
8. The overall idea is not new (agiM);

It seems the rebuttal mainly focuses the concerns from FDp2 and W8Yb, but misses to address the other concerns, including #1, #2, #4, #6 and #8. The AC doesn't think the reviewers will be convinced, and thus suggests to reject this submission. In addition, the rebuttal frequently refers to the wrong concern, which makes AC judgement more difficult. The authors are strongly suggested to improve the presentation and writing.

**Reviewer Concerns:**

However, the reviewer concerns include:
1. The details of BrowseComp-Small is unclear and its comparison with the full set is missing (FxaA, agiM);
2. Missing comparison on other popular benchmarks (FxaA, FDp2);
3. Concerns on statistical rigor and baseline comparison fairness (FDp2, agiM);
4. Limited ablations and missing qualitative analysis (FDp2, agiM);
5. LLM judge dependence (FDp2);
6. Absolute performance still extremely low (FDp2, W8Yb);
7. Missing the performance of different LLMs (W8Yb);
8. The overall idea is not new (agiM);

It seems the rebuttal mainly focuses the concerns from FDp2 and W8Yb, but misses to address the other concerns, including #1, #2, #4, #6 and #8. The AC doesn't think the reviewers will be convinced.

**Reviewer Scores:**

The pre-rebuttal scores are 2 (FxaA), 2 (FDp2), 6 (W8Yb), 2 (agiM). Most of FDp2's concerns were resolved, whose score may be improved (but not necessarily to positive score). But FxaA/agiM's concerns were barely addressed, and I don't think their scores will be improved.

---

### Decision · Program_Chairs · 2026-01-26

Reject